# A Lesion-Wise Cascade for False-Positive Reduction and Lesion-Level Brain Tumor Typing in Multi-Sequence MRI

**Shih-Ting Tai**[1]                                          TINAM710SS@GMAIL.COM
**Yi-Chen Sun**[1]                                           S.CHENNEY0830@GMAIL.COM
**Jing-Jhong Chen**[1]                                       ABC2001925@GMAIL.COM
**Jia-Sheng Hong**[1]                                        SAMHONG511017@GMAIL.COM
**Weir-Chiang You**[2]                                       BIOJOHNYOU@VGHTC.GOV.TW
**Cheng-Chia Lee**[3]                                        CCLEE12@VGHTPE.GOV.TW
**Chih-Chun Wu**[4]                                          CCWU1005@GMAIL.COM
**Hsiu-Mei Wu**[4]                                           HMWU@VGHTPE.GOV.TW
**Hung-Chieh Chen**[5]                                       HCCHEN8@VGHTC.GOV.TW
**Han-Jui Lee**[4]                                           HJLEE2@VGHTPE.GOV.TW
**Yu-Te Wu**[*1,6]                                           YTWU@NYCU.EDU.TW

[1] *Institute of Biophotonics, National Yang Ming Chiao Tung University, Taipei City, Taiwan*

[2] *Department of Radiation Oncology, Taichung Veterans General Hospital, Taichung City, Taiwan*

[3] *Department of Neurosurgery, Neurological Institute, Taipei Veterans General Hospital, Taipei City, Taiwan*

[4] *Department of Radiology, Taipei Veterans General Hospital, Taipei City, Taiwan*

[5] *Department of Radiology, Taichung Veterans General Hospital, Taichung City, Taiwan*

[6] *Center for Smart Health and Medicine, Taipei City Hospital, Taipei City, Taiwan*

## Abstract

In clinically relevant brain tumor detection settings such as stereotactic radiosurgery (SRS), minimizing missed lesions is critical because timely detection may preserve treatment opportunities. To reduce the risk of overlooked lesions, automated detection systems often prioritize high sensitivity; however, this typically leads to a substantial lesion-level false-positive (FP) burden that increases radiologist workload and limits practical utility. We propose a lesion-wise cascade that first suppresses likely non-tumorous mimics and then performs six-class lesion-level classification, assigning each retained detection to either the FP class or one of five tumor subtypes: neuroma, meningioma, metastasis, glioma, and pituitary adenoma. Subtype-aware classification is clinically relevant because the assessment of brain masses requires imaging-based differential diagnosis, even though definitive diagnosis often depends on histopathology. A lesion-wise formulation is also important because patient-level labels may obscure lesion-level heterogeneity. On a temporally held-out independent test cohort with a clinically representative prevalence distribution, the full cascade achieved 90.13% sensitivity, 0.41 false positives per scan (FP/scan), and 66.34% precision. Relative to the raw candidate pool, this corresponded to a 73.2% reduction in FP/scan (1.53 to 0.41) at a cost of 3.59 sensitivity points. For lesion-wise tumor subtype classification, the framework achieved 89.55% five-class accuracy on tumor lesions surviving the cascade. These results suggest that the proposed cascade supports both practical lesion-level FP reduction and clinically relevant subtype-aware classification.

**Keywords:** Brain Tumor, Deep Learning, Multi-Sequence MRI, False-Positive Reduction, Lesion-Level Classification, Brain Tumor Typing

---

[*] Corresponding author

## 1. Introduction

In brain tumor detection workflows relevant to stereotactic radiosurgery (SRS), missed or delayed lesion recognition may affect local treatment opportunities, and longitudinal MRI surveillance imposes substantial lesion-level review workload (Garcia et al., 2018; Derks et al., 2022; Hammer et al., 2024). Automated detection systems therefore prioritize high lesion sensitivity, but this often produces a substantial lesion-level false-positive (FP) burden that limits practical utility (Booth and Boyd-Ellison, 2015; Mordang et al., 2016). Candidate-based cascades are commonly used for FP reduction after sensitive proposal generation in medical imaging, typically by applying a second-stage classifier to suppress mimics or non-lesion candidates (Mordang et al., 2016; Cao et al., 2020; Al-masni et al., 2020; Li et al., 2023). However, the central challenge remains the sensitivity–FP trade-off, where aggressive filtering reduces candidate burden but risks removing true lesions. Moreover, brain-mass assessment requires imaging-based differential diagnosis beyond detection, and patient-level predictions may obscure lesion-level heterogeneity; MRI-based tumor classification can therefore support expert review, although definitive diagnosis often relies on histology (Smirniotopoulos and Jäger, 2020; Coupet et al., 2022; Srinivasan et al., 2024). We therefore propose a lesion-wise cascade that refines a high-sensitivity candidate pool by combining sequential FP suppression with five-class tumor subtype classification.

## 2. Method

The framework converts T1-weighted contrast-enhanced (T1C) and T2-weighted (T2) MRI into lesion-wise decisions through a three-stage cascade (Figure 1).

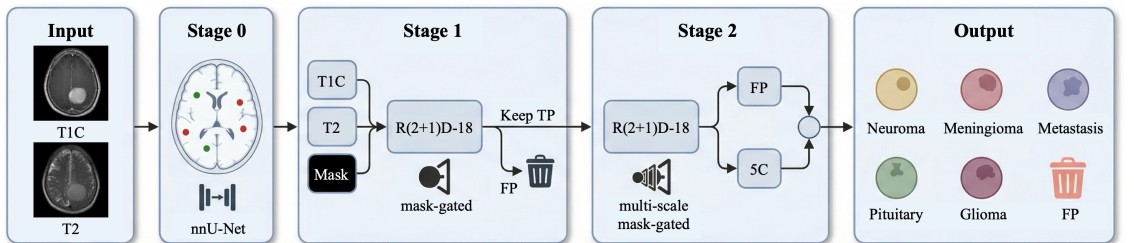

Figure 1: Lesion-wise cascade.

**Stage-0 candidate generation.** We use a 3D nnU-Net v2 sensitivity-oriented proposal generator trained on the same development cohort, with co-registered T1C/T2 MRI and tumor masks. Except for a $64 \times 64 \times 64$ patch size chosen to favor small-lesion sensitivity, training followed the default nnU-Net v2 configuration. Connected components from the predicted mask defined lesion candidates and the downstream sensitivity upper bound.

**Lesion-wise candidate definition and patch construction.** For training, each candidate is matched to the ground-truth lesion by maximum volumetric intersection-over-union (IoU); candidates with IoU $> 0.10$ are labeled true positives (TP), and all others, including tumor-negative candidates, are labeled false positives (FP). Each candidate is represented as a $128 \times 128 \times 15$ XY-full 2.5D patch with T1C, T2, and predicted-mask channels.

**Stage-1 lesion-wise FP reduction.** Stage-1 is a binary TP/FP classifier built on a Kinetics-400-pretrained R(2+1)D-18 backbone (Tran et al., 2018; Kay et al., 2017) with mask gating. It is trained using cross-entropy warm-up followed by asymmetric focal loss to address TP/FP imbalance. Stage-1 performs coarse FP filtering while preserving high lesion sensitivity. During inference, it outputs a TP probability, and candidates below the keep threshold are labeled as FP and excluded from Stage-2.

**Stage-2 final lesion classification.** Candidates retained by Stage-1 are passed to Stage-2, which uses a separate Kinetics-400-pretrained R(2+1)D-18 backbone, a mask-gating module composed of parallel dilated convolutions with multiple dilation rates, and two prediction heads: one for FP-versus-tumor discrimination and one for five-class tumor subtype classification. During training, Stage-2 incorporates prevalence-aware six-class sampling and size scaling. Unlike Stage-1, Stage-2 jointly models residual false positives and tumor subtypes to handle more challenging candidates that survive Stage-1. During inference, candidates with FP probability above the Stage-2 threshold are labeled as FP; the remaining candidates are assigned a tumor subtype by argmax over the five tumor classes.

## 3. Results

**Data and operating-point selection.** Of 12,437 dual-center scans, 12,188 formed the development set, yielding 20,305 Stage-0 patches (12,342 TP/7,963 FP), split case-wise 80/20 without overlap. The test set comprised 249 temporally held-out scans (126 negative/123 positive; 223 lesions) yielding 590 candidates (209 TP/381 FP), leaving 14 fixed false negatives. Thresholds were selected sequentially: Stage-1 keep 0.50, Stage-2 FP 0.85.

**Main results.** Table 1 summarizes the main lesion-wise results. Relative to Stage-0, Stage-1 removed 260 false-positive candidates and 6 true-positive lesions, reducing FP/scan to 0.49 ($-1.04$), lesion sensitivity to 91.03% ($-2.69$ points), and increasing lesion precision to 62.65% ($+27.23$ points). Relative to Stage-0, the full cascade removed 279 false-positive candidates and 8 true-positive lesions, reducing FP/scan to 0.41 ($-1.12$), lesion sensitivity to 90.13% ($-3.59$ points), and increasing lesion precision to 66.34% ($+30.92$ points). Among tumor lesions retained by the full cascade, Stage-2 achieved 89.55% overall five-class classification accuracy, with class-wise sensitivity ranging from 81.08% to 100.00%.

Table 1: Main deployment results on the independent test cohort.

| Variant | Sens. | Prec. | FP/scan | Rem. FP | Rem. TP | Tumor Subtype Acc. |
|---|---|---|---|---|---|---|
| Stage-0 Baseline | 93.72 | 35.42 | 1.53 | – | – | – |
| + Stage-1 | 91.03 | 62.65 | 0.49 | 260 | 6 | – |
| + Full cascade | 90.13 | 66.34 | 0.41 | 279 | 8 | 89.55 |

Subtype accuracy computed on lesions surviving the full cascade; removed FP/TP relative to Stage-0.

**Overall summary.** Stage-1 delivered most of the false-positive reduction, whereas Stage-2 added residual FP refinement together with five-class subtype classification. Overall, the cascade reduced false-positive burden, improved precision, and enabled clinically relevant lesion-level review while maintaining high sensitivity.

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

## Appendix A. Supplementary Material

This appendix summarizes the Stage-0 proposal-generator setup, Stage-1/2 classifier training details, threshold-sweep analyses for operating-point selection, bootstrap confidence intervals for the main deployment comparison, and the final five-class confusion matrix.

Table 2: Stage-0 proposal-generator summary.

| Item | Description |
|---|---|
| Model / role | 3D nnU-Net v2 sensitivity-oriented proposal generator |
| Training cohort | Same development cohort as downstream classifiers; temporal test cohort held out |
| Input / supervision | Co-registered T1C/T2 MRI; binary tumor masks |
| Training settings | Default nnU-Net v2 configuration except $64 \times 64 \times 64$ patch size |
| Candidate definition | Connected components from the predicted mask; defines the candidate pool and downstream sensitivity upper bound |

Table 3: Stage-1 and Stage-2 classifier training summary.

| Component | Stage-1 | Stage-2 |
|---|---|---|
| Patch implementation | XY-full 2.5D patches with 15 axial slices and $128 \times 128$ in-plane resize; T2 and masks aligned to T1C; candidate-centered z-cropping based on the Stage-0 prediction. | |
| Normalization | Robust z-scoring within a 5-mm dilated candidate foreground region of interest (ROI). | |
| Optimization | AdamW (learning rate $1 \times 10^{-3}$, weight decay $1 \times 10^{-2}$), batch size 50, up to 300 epochs with early stopping (patience 30). | |
| Backbone | Kinetics-400-pretrained R(2+1)D-18 backbone. | Kinetics-400-pretrained R(2+1)D-18 backbone. |
| Gate | Mask gating. | Mask-gating module with parallel dilated convolutions at multiple dilation rates. |
| Heads | – | One FP-versus-tumor head and one five-class tumor subtype head. |
| Loss | Cross-entropy warm-up followed by asymmetric focal loss. | Binary cross-entropy for the FP head and focal cross-entropy for the tumor head. |
| Sampling / scaling | Target prevalence 1:1 (FP:TP). | Prevalence-aware six-class sampling $\{50, 9, 18, 21, 16, 6\}$; training-only size scaling. |

Table 4: Stage-1 threshold sweep on the test deployment setting.

| Stage-1 TP threshold | TP | FP | Sensitivity (%) | Precision (%) | FP/scan |
|---|---|---|---|---|---|
| 0.10 | 209 | 349 | 93.72 | 37.46 | 1.40 |
| 0.20 | 209 | 300 | 93.72 | 41.06 | 1.20 |
| 0.30 | 205 | 212 | 91.93 | 49.16 | 0.85 |
| 0.40 | 205 | 139 | 91.93 | 59.59 | 0.56 |
| 0.50 | 203 | 121 | 91.03 | 62.65 | 0.49 |
| 0.60 | 194 | 67 | 87.00 | 74.33 | 0.27 |
| 0.70 | 190 | 48 | 85.20 | 79.83 | 0.19 |
| 0.80 | 190 | 38 | 85.20 | 83.33 | 0.15 |
| 0.90 | 189 | 26 | 84.75 | 87.91 | 0.10 |

Table 5: Stage-2 threshold sweep on the test deployment setting after fixing the Stage-1 TP keep threshold at 0.50.

| Stage-2 FP threshold | TP | FP | Sensitivity (%) | Precision (%) | FP/scan |
|---|---|---|---|---|---|
| 0.15 | 186 | 73 | 83.41 | 71.81 | 0.29 |
| 0.20 | 187 | 74 | 83.86 | 71.65 | 0.30 |
| 0.25 | 187 | 76 | 83.86 | 71.10 | 0.31 |
| 0.30 | 190 | 80 | 85.20 | 70.37 | 0.32 |
| 0.40 | 193 | 89 | 86.55 | 68.44 | 0.36 |
| 0.50 | 194 | 93 | 87.00 | 67.60 | 0.37 |
| 0.60 | 196 | 98 | 87.89 | 66.67 | 0.39 |
| 0.70 | 197 | 98 | 88.34 | 66.78 | 0.39 |
| 0.80 | 199 | 100 | 89.24 | 66.56 | 0.40 |
| 0.85 | 201 | 102 | 90.13 | 66.34 | 0.41 |
| 0.90 | 201 | 106 | 90.13 | 65.47 | 0.43 |

Table 6: Bootstrap 95% confidence intervals for the main deployment comparison, estimated from 2,000 case-level bootstrap resamples.

| Pipeline Variant | Sensitivity (%) | Precision (%) | FP/scan |
|---|---|---|---|
| Stage-0 Baseline | 93.72 [92.13, 94.89] | 35.42 [29.37, 41.87] | 1.53 [1.34, 1.73] |
| + Stage-1 | 91.03 [87.78, 93.41] | 62.65 [54.45, 70.38] | 0.49 [0.39, 0.59] |
| + Full cascade | 90.13 [86.52, 92.86] | 66.34 [58.30, 73.75] | 0.41 [0.32, 0.50] |

Table 7: Five-class confusion matrix for tumor lesions surviving the full cascade.

| Ground Truth \ Predicted | Neu | Men | Met | Gli | Pit | Total | Sensitivity (%) |
|---|---|---|---|---|---|---|---|
| Neuroma | 16 | 0 | 1 | 1 | 0 | 18 | 88.89 |
| Meningioma | 1 | 30 | 3 | 3 | 0 | 37 | 81.08 |
| Metastasis | 0 | 2 | 107 | 10 | 0 | 119 | 89.92 |
| Glioma | 0 | 0 | 0 | 16 | 0 | 16 | 100.00 |
| Pituitary adenoma | 0 | 0 | 0 | 0 | 11 | 11 | 100.00 |
| Total | 17 | 32 | 111 | 30 | 11 | 201 | 89.55 |

Neu, neuroma; Men, meningioma; Met, metastasis; Gli, glioma; Pit, pituitary adenoma.

