# OpenReview forum: "A Lesion-Wise Cascade for False-Positive Reduction and Lesion-Level Brain Tumor Typing in Multi-Sequence MRI"
_MIDL.io/2026/Short_Papers — MIDL 2026 - Short Papers Poster_

### Official Review · Reviewer_ZpyZ · 2026-05-03
**Well motivated lesion based FP reduction**

**Rating:** 4
**Confidence:** 5

**Review:**

This paper is generally well written and addresses the clinically relevant issue of high false-positive burden in sensitivity-oriented tumor detection pipelines. The proposed lesion-wise cascade pipeline is relevant and achieves a meaningful reduction in FP/scan with only a modest loss in sensitivity. The general idea of FP reduction after a sensitive detection has already been quite explored in the literature. However, the idea of prposing a lesion-based reduction coupled with lesion subtype classification is quite interesting and can help add more context to the model, thereby improving FP reduction while retaining TP. The main limitations are the limited description of the Stage‑0 nnU-Net detector, which largely determines the initial FP burden, and the minimal discussion of closely related FP-reduction approaches, for which additional citations or brief comparisons would strengthen the paper despite space constraints.

**Summary:**

The authors propose a three-stage lesion-wise cascade to improve the precision of a lesion-detection algorithm and to jointly classify tumors in stereotactic radiosurgery workflows. The pipeline uses an initial sensitive nnU-Net for candidate generation, followed by a binary classifier for performing a first step of FP filtering, and a final head for joint FP refinement and five-class tumor classification. On a held-out test set, the framework reduced FP/scan from 1.53 to 0.41/scan while maintaining 90.13% sensitivity.

**Strengths:**

- The clinical motivation of the work is indeed important.
- The proposed pipeline, although not novel, remains quite clear.
- Good FP reduction rate with modest sensitivity loss.
- Inclusion of tumor subtype classification adds more context to the model.
- The paper is well written and easy to follow.

**Weaknesses:**

- Limited discussion and citation of prior FP-reduction cascades.
- Stage‑0 detector is merely described (training set, model, loss, etc) despite its strong impact on the whole study results.
- Final FP rate (0.41 FP/scan) may still be non-negligible in practice, but it is a good start.
- No ablation analysis, it is understandable with the short paper track.

**Justification Of Rating:**

Overall, the idea presented in the paper is well motivated and demonstrates a ùeaningful reduction in false positives. The contribution needs to be more ablated and more detailed, but remains appropriate for the short paper track.

---

### Decision · Program_Chairs · 2026-05-08

Accept (Poster)